# Detection of Mild Cognitive Impairment with MEG Functional Connectivity Using Wavelet-Based Neuromarkers

**DOI:** 10.3390/s21186210

**Published:** 2021-09-16

**Authors:** Su Yang, Jose Miguel Sanchez Bornot, Ricardo Bruña Fernandez, Farzin Deravi, Sanaul Hoque, KongFatt Wong-Lin, Girijesh Prasad

**Affiliations:** 1Department of Computer Science, Swansea University, Swansea SA1 8EN, UK; 2Intelligent Systems Research Centre, School of Computing, Engineering and Intelligent Systems, Ulster University, Londonderry BT48 7JL, Ireland; jm.sanchez-bornot@ulster.ac.uk (J.M.S.B.); K.Wong-Lin@ulster.ac.uk (K.W.-L.); G.Prasad@ulster.ac.uk (G.P.); 3Centre for Biomedical Technology, Technical University of Madrid, 28223 Madrid, Spain; Ricardo.Bruna@ctb.upm.es; 4School of Engineering, University of Kent, Canterbury CT2 7NZ, UK; F.Deravi@kent.ac.uk (F.D.); S.Hoque@kent.ac.uk (S.H.)

**Keywords:** MEG, connectivity coherence, wavelet-based neuromarker, hyperconnectivity, hypoconnectivity, MCI detection

## Abstract

Studies on developing effective neuromarkers based on magnetoencephalographic (MEG) signals have been drawing increasing attention in the neuroscience community. This study explores the idea of using source-based magnitude-squared spectral coherence as a spatial indicator for effective regions of interest (ROIs) localization, subsequently discriminating the participants with mild cognitive impairment (MCI) from a group of age-matched healthy control (HC) elderly participants. We found that the cortical regions could be divided into two distinctive groups based on their coherence indices. Compared to HC, some ROIs showed increased connectivity (hyper-connected ROIs) for MCI participants, whereas the remaining ROIs demonstrated reduced connectivity (hypo-connected ROIs). Based on these findings, a series of wavelet-based source-level neuromarkers for MCI detection are proposed and explored, with respect to the two distinctive ROI groups. It was found that the neuromarkers extracted from the hyper-connected ROIs performed significantly better for MCI detection than those from the hypo-connected ROIs. The neuromarkers were classified using support vector machine (SVM) and k-NN classifiers and evaluated through Monte Carlo cross-validation. An average recognition rate of 93.83% was obtained using source-reconstructed signals from the hyper-connected ROI group. To better conform to clinical practice settings, a leave-one-out cross-validation (LOOCV) approach was also employed to ensure that the data for testing was from a participant that the classifier has never seen. Using LOOCV, we found the best average classification accuracy was reduced to 83.80% using the same set of neuromarkers obtained from the ROI group with functional hyper-connections. This performance surpassed the results reported using wavelet-based features by approximately 15%. Overall, our work suggests that (1) certain ROIs are particularly effective for MCI detection, especially when multi-resolution wavelet biomarkers are employed for such diagnosis; (2) there exists a significant performance difference in system evaluation between research-based experimental design and clinically accepted evaluation standards.

## 1. Introduction

With the recent surge in the development of machine learning (ML) techniques, particularly the success of machine learning in speech and image recognition [1,2], ML-aided clinical diagnosis has also received increasing attention in the healthcare research community [3]. Due to extreme complexity and relatively limited understanding of the human brain, it is quite expensive to train clinicians to gain ample experience in diagnosing brain malfunctions. In addition, due to geographical and national boundaries, such valuable expertise is restricted and not universally available. On the other hand, compared to the case for a human being, the required time to train an ML-based expert system for diagnosis is much shorter. Moreover, technically, the experience gained by such an expert system is portable, which may be easily transferred to where it is needed and retained for future use. These appealing features have made researchers put increasing effort into developing advanced ML algorithms for the detection of brain malfunctions.

As a result of continuing improvements in healthcare provision, life expectancy has been increasing in many countries. Consequently, the prevalence of Alzheimer’s disease (AD) has also been rising. AD is one of the brain malfunctions whose exact triggering reasons are still not fully established [4], and its early prediction and detection has become an active research topic. A timely diagnosis of mild cognitive impairment (MCI), as the prodromal stage of AD, is therefore vital to preventing or delaying the development of AD. Magnetoencephalography (MEG) is an effective neuroimaging modality that has been increasingly drawing attention in the field of ML-aided MCI detection [5]. Compared to other conventional modalities such as electroencephalography (EEG) and magnetic resonance imaging (MRI), MEG arguably benefits from both higher temporal resolution (vs. fMRI) and better spatial resolution (vs. EEG) [5].

The major signal processing phases, such as signal pre-processing, neuromarker (feature) extraction, and feature classification/recognition, should be carefully engineered while developing an ML-aided diagnosis system. Of all these important steps, the development of effective neuromarkers is possibly one of the most important tasks for a MEG-based MCI detection system, not only from the ML point of view but also from the clinical perspective. A good neuromarker developed based on Explainable AI should not only provide decent classification performance, but also reveal useful characteristics that reveal the changes accompanying a particular cognitive disorder, e.g., where the anatomical origins of MCI alterations at locations that MCI evidence stems from.

A number of ML-aided diagnosis approaches using MEG source-level analyses for MCI detection have been reported recently [5]. For example, Nakamura et al. [6] investigated the prodromal stages of AD using source-space MEG signals reconstructed through the linearly constrained minimum variance (LCMV) beamforming technique. It involved using MEG data obtained from 28 individuals with MCI and 38 cognitively normal individuals for feature extraction. The preliminary feature extracted was the regional spectral patterns which were then integrated with information acquired from Pittsburgh compound PiB-PET, fluorodeoxyglucose-PET, structural MRI, and outcomes of cognitive tests. The results demonstrated that regional spectral patterns of resting-state activity could be separated into several types of MEG signatures, which may be used as useful biomarkers to study the pre-dementia stages of AD [6]. A related experiment conducted by Jacini et al. [7] also used LCMV beamforming for MEG signal reconstruction, with an aim to detect amnestic Mild Cognitive Impairment (aMCI) from healthy controls (HC). Significant differences between the two groups were found in both temporal poles, which are associated with memory processing.

Dimitriadis et al. [8] recently proposed a number of analysis protocols based on the source-reconstructed MEG signals. Different analytic strategies for single-layer and multi-layer representations of functional brain networks were explored. Three connectivity estimators, namely phase-locking value (PLV), the imaginary part of PLV, and the correlation of the envelope (CorrEnv) were computed and used as neuromarkers. Four minutes of resting-state activity was obtained from 24 MCI patients and 30 healthy controls, using a 306-channel Elekta Vectorview system [9]. The source reconstruction was performed using an LCMV beamformer. Both the intra- and cross-frequency interactions were estimated by the three connectivity estimators within the seven selected frequency bands. The highest classification accuracy of 98% was reported in this study using the CorrEnv estimator based on a 5-fold cross-validation, while 70% accuracy was reported for leave-one-out cross-validation using the same brain regions and features.

MEG-based MCI detection using time-frequency analysis has also been explored by Poza et al. [10]. Following a wavelet transform of the signals from 148 channels, the wavelet turbulences were measured and used as the features for MCI vs. HC classification. A number of turbulence coefficients, which reflect the correlations between location, spread, and shape of the wavelet domain signals, were computed. ROC analysis was employed to demonstrate the detection performance. Their experiments involved 18 MCI subjects and 27 controls, the highest detection rate of 68.9% was reported by following the leave-one-out cross-validation procedure.

Considered as non-stationary measurements [11], both EEG and MEG signals share certain similarities in their dynamic characteristics. However, using wavelets for feature extraction based on EEG has received greater attention due to the wider availability of EEG systems. For instance, Fiscon et al. [12] proposed employing discrete wavelet transform for the preliminary feature analysis of EEG data; the resulting coefficients were used to compute a number of statistics, including mean, standard deviation, and power spectral density for each segmented signal. Three minutes of recording captured by a 19-electrode EEG cap was used to classify two groups: 23 HC and 37 MCI subjects. A recognition rate of 92% was reported using a decision tree classifier. Among the wavelet-based feature extraction approaches, this was considered a superior recognition performance compared to using MEG for MCI detection.

As seen in the aforementioned studies, the range of ML techniques that have been used have shown high variability in MCI detection performance. In this work, we conducted a series of experimental evaluations using MEG data for MCI vs. HC classification, in order to establish which of these evaluation metrics could reflect real-world scenarios. The study reported in this paper covers the following four main aspects:(1)A regional analysis of the effectiveness of different cortical regions for MCI detection is undertaken to examine the phenomena of hyper- and hypo-connections between regions of interest (ROIs) which may have a significant impact on MCI vs. HC classification with respect to ROI selections.(2)Based on the findings from the regional analysis, a system for MCI/HC classification is developed. In particular, the effectiveness of the novel wavelet-based neuromarkers for MCI detection is evaluated. The results show that the proposed method can achieve an average classification accuracy of about 84%, using only MEG data from the selected cortical regions.(3)A comparative analysis using two different evaluation procedures is undertaken. One is based on the well-known Monte Carlo repetition of random splits, i.e., into 95/5% of training/test subsets. The other test procedure is a subject-based 20-fold leave-one-out (LOO) cross-validation, in order to better mimic the actual clinical scenario with individual participants (each class contained 20 subjects).(4)The region selection and recognition problems are addressed from a practical perspective by exploring the statistical significance of the results, which is a crucial aspect that is often ignored in the performance evaluation for such recognition systems.

The rest of the paper is organized as follows: In Section 2, a regional analysis of using MEG for MCI detection is conducted. The selected ROIs are then used for classification, based on magnitude-squared spectral coherence measurements in the source space. Section 3 presents the approach of using wavelet-based neuromarker derived from different ROI groups for MCI detection; this exploration also serves as a justification for the regional preferences acquired from the regional analysis in the previous section. In Section 4, a series of statistical evaluations are conducted to better justify the significance of the findings from this study. Section 4 contains a discussion and conclusion, along with a few suggestions for future work.

## 2. Region of Interest Analysis

In this section, an objective approach is presented for finding a subset of ROIs, which may be relatively more effective for MCI detection. The analysis is based on a neuroimaging database [13] obtained from the Hospital Universitario de San Carlos (Madrid, Spain). This database contains MEG recordings from 20 MCI and 20 HC participants collected using an Elekta Neuromag MEG 306-sensor recording system [9]. The data of each participant comprises three minutes of usable recording in the resting state while participants’ eyes were closed. All participants were right-handed subjects as verified with the Edinburgh Handedness Inventory (Oldfield, 1971) [14]. The channels that captured blinks and eye movement signals were excluded from the analysis in this work. The data were processed offline using the temporal extension of the signal space separation algorithm [15] using MaxFilter 2.2 (provided by Elekta) with a window length of 10 s and a correlation threshold of 0.9.

For the source-level analysis, a sphere-based forward model was employed and was realigned with the individual’s MRI in the sensor space. The minimum norm estimation (MNE) technique was adopted for inverse modeling [16]. One important reason that the MNE has been adopted for source signal reconstruction, rather than the spatial filters (such as beamformer) is that MNE has been found to be effective at coupling between the point-like sources and in reflecting the interacting pairs of extended patch coherences [17]. In this work, we propose to employ coherence as the connectivity indicator between the sources. Only the data from magnetometers (102 sensors in total) were subsequently transformed into a source-level matrix, which covered 8196 vertices of the cortical surface. This is considered as an appropriate number for source placements to balance the analysis resolution and source space dimensionality [18]. The cortical surface was divided into left and right hemispheres, each hemisphere was then further divided into five conventional regions [19].

The intra-hemispheric and inter-hemispheric connectivity were also explored. The investigations addressed the problem of ROIs selection from three perspectives: (1) computation of connectivity indicator; (2) analysis of single ROI vs. rest of the regions; (3) ROI sensitivity analysis for MCI detection. The follow-up subsections are devoted to the discussion of the connectivity indicator (Section 2.1), analysis methodology (Section 2.2), and statistical tests of the proposed region selection approach (Section 2.3).

### 2.1. Connectivity Indicator Computation

The aim of this section is to statistically evaluate and decide the region(s) that may be relatively more effective in ML-aided MCI diagnoses. Given the issues that typically affect source localization solutions, (i.e., volume conduction and field spread, or the blurriness of estimated sources), we decided to group the source space into ten conventional surface regions within the left/right hemisphere to study their possible interactions. In detail, the cortical surface is partitioned into frontal (caudal middle frontal gyrus, frontal pole cortex, lateral orbitofrontal cortex, pars triangularis gyrus, rostral middle frontal gyrus, superior frontal gyrus), temporal (middle temporal gyrus, pars orbitalis gyrus, superior temporal gyrus, temporal pole cortex, inferior temporal gyrus, insular cortex), parietal (inferior parietal cortex, superior parietal gyrus, supramarginal gyrus), occipital (lateral occipital cortex), and central (postcentral precentral gyrus) regions. Another reason for choosing these partitions is the differentiated role(s) of these regions in AD/MCI pathology [20]. In order to explore the functional connectivity of inter and intra hemispheres, during the source reconstruction we purposely excluded the data from six magnetometers (in the sensor space) that are located along the central line of the cortical surface, as the focus of this work is the interactions/differences between the two hemispheres.

After source-level reconstruction, data from the selected 10 regions were averaged and used to represent each region, which provided 10 parcels in the source space. It is worth mentioning that 3 min of recording was used for each participant. For connectivity analysis, we windowed the data into 180 one-second epochs, with each epoch being comprised of 200 samples (data was down-sampled to 200 Hz from 1000 Hz). Therefore, in total, 3600 epochs (20 × 180) were generated for each class per ROI. Each epoch (which is the mean of the parcels from its corresponding region) was then paired with every other epoch from other regions for the coherence analysis, and their pair-wise functional connectivity was estimated subsequently. As a result, we were able to estimate the pairwise functional connectivity (FCs) among the ten cortical regions, as illustrated in Figure 1.

The broadband methods for functional connectivity analysis have been found to be effective for EEG signals for seizure detection [23]. In our study, it is found that concatenating individual MEG frequency bands (delta, theta, alpha, beta, and gamma, 0.1~80 Hz) can boost the MCI detection performance. The performance of the effective bands after wavelet decomposition is included in the later sections. When all of the bands were used, the highest classification performance was achieved. In this study, the resting state data without particular band-sensitive stimulus were collected for region selection purposes. The broadband (0.5 to 80 Hz) spectral coherence between two signals after source reconstruction was employed as a measurement of the functional connectivity between the ROIs. The spectral coherence between two signals can be defined as follows [24]:(1)$$C_{xy}\left(f\right)=\frac{{\left|P_{xy}\left(f\right)\right|}^{2}}{P_{xx}\left(f\right)P_{yy}\left(f\right)}$$
(2)$$P_{xy}\left(f\right)=\overset{\infty }{\underset{m=-\infty }{\sum }}R_{xy}\left(m\right)e^{-j\omega m}$$
where $\;P_{xy}\left(f\right)$ is the cross-spectral density between signals *x* and *y*, $P_{xx}\left(f\right)$ and $P_{yy}\left(f\right)$ are the auto-power spectral density of *x* and *y*, respectively. $R_{xy}\left(m\right)$ defines the cross-correlation sequence, which can be expressed as: $E\left\{x_{n+m}y_{n}^{*}\right\}$, in which $x_{n}$ and $y_{n}$ are jointly stationary random processes, −∞< *n* <∞, −∞< *n* <∞, and E{∗} is the expected value operator. If the coherence coefficients between two signals approach 1, it indicates strong connectivity; while the coherence coefficients approaching 0 indicate that there is no connectivity between the two. Here, we compute the spectral coherence between all possible pairs of the *f* selected ROIs.

### 2.2. Pair-Wise ROI Analysis

To leverage the imaginary part of coherence, and to mitigate the possible volume conduction effect in MEG signals while computing the absolute value of the coherence [25], we computed the connectivity strength for each region with respect to the rest of the regions using the magnitude-squared coherence (MS-COH) as shown in (1). This is achieved as follows: given an averaged parcel from one region, its MS-COHs for all possible regional pairs were first computed, and their sum was used to indicate the functional connectivity (FC) strength of the region. The connectivity values for each ROI region are included in Table 1. The strength of every region can then be defined by the sum of all the FCs resulting from interactions between itself and other regions [26].

The results shown in Table 1 are the averaged MS-COHs data from 20 participants for each class (i.e., HC and MCI separately). For the selected ten ROIs, each one was paired with the other nine ROIs for coherence computations. For example, LF indicated the ROI pair between the left frontal lobe and the remaining regions. The summation of the coherences with the other regions could also be viewed as the degree of connectivity for the LF region. Note that the data shown in Table 1 are the averaged values of 3600 summed MS-COHs for each class (20 × 180, 20 participants per class, 180 epochs per participant).

The detailed steps required for this computation are shown as follows:(1)Each ROI contained hundreds of MEG values in the source space. First, the values of each ROI were averaged: this resulted in ten mean MEG values for the ten selected ROIs.(2)For each averaged MEG value per ROI, the coherences with respect to the remaining nine mean ROI values were computed, which produced nine coherences for each ROI.(3)The nine coherence values for each ROI were summed together and the resulting summation is defined as the degree of connectivity for each ROI.(4)Considering that there are 20 participants per class and 180 s per participant, 20 × 180 = 3600 epoch per ROI for each class are produced.(5)The max, min, mean, and SD for these 3600 epochs, with respect to the degree of connectivity for each ROI, are computed and shown in Table 1.

Using the FCs of HC participants as the baseline, the averaged FCs of eight regions were found to be reduced for MCI subjects, except for the left frontal (LF) and right central (RC) regions with increased FCs. Further exploring the statistics of MS-COHs in Table 1, it was noticed that the MS-COHs for HC participants generally had larger ranges of distribution/variance for most of the regions, except the left frontal lobe (LF), which had increased the standard deviation of FC in MCI participants.

The data in Table 1 suggest that, compared to HC participants, FC between LF and RC regions was slightly increased for MCI participants, while the fact that FCs for all the remaining regions decreased in the MCI class. The differences in FC observed for different ROIs in this study are largely in line with a few previous findings in the literature [13,27,28,29]. For example, hyper synchronization has been observed in MCI with two significant networks, involving frontotemporal and fronto-occipital connections. A diminished functional connectivity ratio was reflected in the progressive MCI group [13]. In [28], connectivity was found to be increased in APOE-e4 carriers in a set of mostly right-hemisphere connections, including lateral parietal and precuneus regions of the Default Mode Network. It is, therefore, justified to further explore whether the change of FC could also be effectively used as indicators to distinguish MCI from HC.

It should be noted that the degree of connectivity per region (the summation of nine regional MS-COHs) as suggested by the data in Table 1, does not indicate the strength of particular connections between pairs of regions; it reflects only the overall strength of connection to each region. To further identify the regions which are contributing to increasing the connection strength for the LF and RC regions, an exhaustive analysis was conducted to explore the connectivity of all possible ROI pairs. There were 10 ROIs selected in this study, therefore, in total, 45 unique ROI combination pairs were needed for estimating the non-directional FCs per class.

As is shown in Figure 2, all of the 45 region-pair combinations were illustrated for both the HC and MCI participants in terms of the averaged MS-COHs. Compared to the HC group, we noticed three regional combinations, namely LF-LT, RT-RC, and LP-LO pairs, revealed hyper-connectivity in MCI participants. These pairs related to the six regions based on our cortical surface partitions, five of them were highlighted in red in Figure 1. The blue regions in Figure 1 thus form the other group of ROIs, which experienced a reduction in FCs. Note that, in this study, the LO lobe was considered as the ROI with hypo-connectivity, due to its lower FC the decrease in FC with respect to the rest of ROIs (as shown in Table 1). Considering LO as a hypo-connectivity region was also to maintain the balance of data volume for objective comparison in the training process (guarantee five ROIs for each group in classification). In addition, it was noticed from Figure 2 that the LT-LC pair had strong connectivity for both HC and MCI. This might be due to these two regions being spatially proximal along the cortical surface.

Further exploring the results in Table 1 and Figure 2, compared to the HC class, the reduction in FC is observed in the majority of ROI regional pairs. This evidence is in line with published reports claiming that both AD and MCI may be due to the disconnection syndromes of the brain [30].

Based on the above analysis, we propose two hypotheses: (1) ROIs showing hypo-connectivity may be reflecting the underlying anatomical changes due to progression towards AD; (2) hyper-connectivity of ROIs may be induced by some compensatory mechanism limited to adjacent areas in MCI groups. It would be interesting to know which group, (i.e., hyper- or hypo-connected regions) will be more sensitive to the MCI, hence to better facilitate the MCI vs. HC classification problem. In particular, the second hypothesis will be addressed in Section 3, as it deserves special attention.

### 2.3. Statistical Analysis for the Connectivity Indicator

The database we explored comprised 40 subjects. To justify its representativeness, we examined the statistical significance of the connectivity analysis results obtained in the previous section. Certain differences in the degree of the regional connections (Table 1) and strengths of the ROIs were noticed (Figure 2). To measure the confidence of these differences, the two-sample *t*-test [31] was used to explore the confidence level for the equality of the means in any two ROI-based coherence sets. 

The *t*-test assumes data samples follow a normal distribution [32]. Given there are 10 ROIs to be explored, histograms of MS-COHs from 45 region pair combinations were exhaustively computed, and all the histograms with respect to the coherences for these regional pairs were analyzed. Though the right tails of some distributions have been found relatively larger than the other side of the histogram, they still can moderately be considered as normally distributed. In addition, for all the $3.24\times 10^{5}$ MS-COHs, divided by class and ROI (i.e., 180 × 40 × 45 values, 180 epochs, 40 subjects, 45 combinations), we further employed the popular Anderson–Darling test to statistically check whether the samples are indeed normally distributed [33,34]. The results indicated that the tests reject the null hypothesis at the 5% significance level for about 96% of all the MS-COHs. Based on this, and that considerably large data samples (MS-COHs) were used for the tests, the *t*-test should therefore provide reliable evaluation [32].

Based on the above results, the statistical significance of the coherence connectivity indicators for HC and MCI classes were further explored separately as well as the connectivity of ROIs for both inter- and intra-hemispheres. It was found that out of 45 ROI pairs, 25 belong to the inter-hemispheric group while the remaining 20 pairs were from the intra-hemispheric group. The standard two-sample *t*-test [31] was performed for inter- and intra-hemispheric groups, with respect to HC and MCI classes. For the HC class, it was found that 89% of the inter-hemispheric connections had α < 0.05, while 87% of the intra-hemispheric connections had α < 0.05. For MCI class, 90% of the inter-hemisphere connections are found with α < 0.05; 87% of the intra-lobe connections are found with α < 0.05. These statistics indicate that the majority of COHs were statistically significant.

To further validate the hypotheses proposed in Section 2.2, in the next section, we propose to use the averaged regional parcels of source-reconstructed signals to derive wavelet-based neuromarkers for HC vs. MCI classification. A series of tests were conducted to quantitatively explore the impact of region selection for MCI detection based on the two typical groups of ROIs explored in Section 2: one group represented COH-based hyperconnected FCs and the other hypoconnected FCs.

## 3. Methodology for MCI Detection

The classification method is firstly introduced in Section 3.1, followed by the results of HC vs. MCI classification using wavelet-based neuromarkers in Section 3.2. In Section 3.3, a series of statistical analyses are presented to support the significance of the results obtained in Section 3.2, along with some optimization of the parameters for the MCI detection system.

### 3.1. Pipeline of the MCI Detection System

Thanks to its advantageous characteristics in time-frequency analysis, WT has also been frequently used in EEG data analysis [35,36]. However, the use of wavelets in resting-state MEG data is yet to be thoroughly explored. In this work, features (neuromarkers) based on wavelet coefficients for MCI detection have been developed and evaluated using both the Monte Carlo repetition (MCR) of random data splitting cross-validation and the leave-one-out cross-validation (LOO) approaches [37]. The receiver operating characteristic (ROC) analysis for both approaches and two ROI groups are used to determine their classification accuracies [38].

Following the regional analysis in the previous section, three minutes of available raw MEG signals were first transformed into source space using the MNE method. The mean values of the ten cortical ROIs were computed, which subsequently produced 10 arrays of averaged source recordings. As illustrated in Figure 3, a segmentation phase was included to divide the signals into 18 epochs, each epoch lasting for 10 s. These epochs were then used for feature extraction.

The first step of the feature extraction was to perform the wavelet packet decomposition (WPD) for each epoch [39]. The Daubechies 4 wavelet function [35] was used to decompose the windowed signal up to level 3. Formally, the time-domain signal x(t) was first mapped into the wavelet space with a scale a and a shift b. The transformation process could then be expressed as follows [39]:(3)$$WT_{\psi }\left\{x\right\}=\langle x,\;\psi_{a,b}\rangle =\int_{-\infty }^{+\infty }x\left(t\right)\cdot \psi_{a,b}\left(t\right)dt$$
where $\;\psi_{a,b}\left(t\right)$ is the scaled and shifted version of a given wavelet function:(4)$$\psi_{a,b}\left(t\right)=\frac{1}{\sqrt{a}}\;\psi \left(\frac{t-b}{a}\right)$$

The wavelet coefficients, $WT_{\psi }\left\{x\right\}$ in (3) revealed both the time and the frequency properties of the signal, which will be used for further feature extraction.

For the sake of completeness, the experimental steps in sequential order are presented as follows:(1)For the source space MEG data, in total, 180 s of the recording was preserved, which was then further segmented into a number of windows in the time domain.(2)The epochs (10 s per epoch) were then down-sampled to 200 Hz, followed by a band-pass filtering process extracting the signal between 0.1 Hz to 100 Hz, which cover all the typical bands (Delta, Theta, Alpha, Beta, and Gamma) using a 4th-order Butterworth filter. Here, we purposely included the frequency content that was higher than 60 Hz, as in principle the MEG modality may retain useful information in the high-frequency range (high gamma band).(3)The wavelet packet decomposition was performed on the resulting time-domain signals. WPD was performed up to level 3; each level covers the selected full frequency range, which facilitates the multi-resolution analysis of the signal in question.(4)The approximate derivative [40,41] was computed on the resulting wavelet coefficients ($WT_{\psi }\left\{x\right\}$ in (3)), which calculates differences between adjacent elements of the wavelet coefficients along the first array dimension.(5)Finally, the multi-dimensional feature vector was constructed by concatenating the standard deviations (SDs) of the obtained approximate derivative coefficients from each band.(6)A range of classifiers that have shown to be effective for MEG/EEG signal classifications were employed to investigate the effectiveness of the features, including k-nearest neighbor (k-NN), linear discriminate analysis (LDA), and support vector machine (SVM).

It is worth noting that according to our preliminary exploration, the classifiers based on the neural network are not effective for MCI detection in this work. One possible reason could be the available data for training is not large enough to converge the training process. The motivation for inserting the Approx-Deriv into the wavelet algorithm was to capture/reflect the nonstationary characteristics of the brain signals [42]. Computation of the SD of the resulting coefficients after the Approx-Deriv step was to reduce the dimensionality of the feature vectors. By keeping all the bands from each wavelet decomposition level, a 14-dimensional feature vector for each averaged parcel was generated (two dimensions for Level-1, four dimensions for Level-2, and eight dimensions for Level-3). Each element of the feature vector was the SD of its corresponding approximate derivative of the wavelet coefficients. Each epoch after segmentation generated one feature vector, and for every subject, an 18 × 14 feature matrix was thus produced.

### 3.2. Experimental Analysis on Classification

In this section, the classification performance of the proposed wavelet neuromarkers is explored using both the MCR and LOO evaluation approaches. The two ROI groups proposed in Section 2, namely the COH hyper-connection and hypo-connection groups, with respect to MCI subjects, are hereby referred to as hyper-COH and hypo-COH groups in this context. Results of using the source-level wavelet features from these two groups for MCI detection are presented in the following sections.

The well-known MCR cross-validation was first employed for performance evaluation. For each class (20 × 18 = 360 observations, 20 subjects), 95% of the data (342 observations) was randomly selected for training the classifier, while the remaining 5% (18 observations) was used to test for performance. This process was repeated 20 times in order to guarantee: (1) the average accuracy was principally representing the true performance of the system, (2) in line with the subject-based LOOCV tests for 20 participants, which was repeated 20 times to exhaustively test all the subjects, each for once only. The 3-NN classifier provided the highest mean recognition rate amongst the three classifiers (k-NN, LDA, and SVM with a 2nd-order polynomial kernel). It is worth noting that the employed classifiers are the conventional algorithms for benchmarking and have demonstrated good performances in previous MEG/EEG classification studies [5,43].

As shown in Figure 4, using the MCR approach, the hyper-COH group of ROIs provided slightly better recognition performance than the ROIs from the hypo-COH group. It is worth mentioning that a few ROI pairs such as the LO-RO (see Figure 2) also demonstrated a small increase in COH connectivity (~0.001). Such minor increases in FC were, however, often accompanied by the global FC reduction observed between ROIs. For example, by inspecting the degrees of FC for these ROIs (see Table 1), features from both LO and RO regions were found with reduced FC. Therefore, for the sake of fair comparison between the two ROI groups, both LO and RO regions were assigned to the hypo-COH group in this study.

To further investigate the impact of the ROI groups on MCI detection, a subject-based LOO cross-validation approach was adopted for imitating a real-world scenario, in which, for each test, the data of a participant that the classifier had never seen was, in turn, used for validation [44]. The remaining data from 19 subjects from each class was used to train the classifier. Unlike the previous MCR cross-validation approach where the remaining data was randomly selected for testing, the LOO cross-validation procedure for this setting guarantees that there is no possibility of using the data from the same participant for both training and testing. The process was repeated 20 times to fully use the data of all participants without overlap. The mean accuracy was then calculated. 

Figure 5 illustrates the performance using LOO cross-validation. We found that SVM, with Euclidean proximity mapping (kernel), provided the highest classification performance for both groups. Compared to that in Figure 4, the recognition rates for both groups were clearly lower in Figure 5 with larger variances. Nevertheless, for the given experimental condition, the hyper-COH ROI group still provided an acceptable performance. The hypo-COH group, however, had clearly shown much worse performance, which may be indicative of these regions does not contribute much to MCI detection. A possible explanation for this could be that the hypo-connectivity for MCI is a global phenomenon of the brain malfunctioning; however, a small proportion of the cortical regions are still able to maintain or even slightly increase the functional connection strength in order to balance the overall connectivity traffic and sustain normal operations of the brain. Such regions have stronger connections to the regions to which they are connected in MCI participants, and are, therefore, more apt to exhibit discriminating patterns compared to the corresponding regions from the HC group. Based on these findings, the increased connectivity for MCI, i.e., the neuromarker(s) derived from the hyper-COH ROI group, may thus possess more value for MCI detection. 

To further explore the effectiveness of the proposed method, Table 2 highlights the mean sensitivities and specificities of the two distinctive groups for the two evaluation approaches used. In particular, we found that with respect to MCI detection, the mean sensitivity of the hypo-COH group with LOO was quite low but the specificity was rather high, which might indicate that the features (from both classes) extracted from the hypo-COH ROI group were more inclined to the HC class.

### 3.3. Statistical Analysis of the Neuromarkers for Classification

In this section, we further conduct a number of statistical evaluations of the wavelet neuromarkers proposed in Section 3.2. For the classification features, we noticed that not all the wavelet features follow the normal distribution. As an example, Figure 6 shows two typical feature histogram distributions: the features from 40–80 Hz in the wavelet domain largely follow a normal distribution, whereas the distribution of features from 0–20 Hz appears to contain two Gaussian mixtures. Due to the mixed nature of the feature distribution, two statistical tests, namely the two-sample *t*-test [31] and Kruskal–Wallis test (which does not assume a normal distribution of the samples in question [45]) were employed for comparison and to better justify the statistical significance of the classification results obtained in Section 3.2. 

The wavelet feature vectors proposed in this study contain fourteen dimensions. The *t*-test and Kruskal–Wallis test were conducted independently for each dimension between the HC and MCI classes. As a rule of thumb, we set α = 0.05 for both types of tests, the resulting *p*-values are shown in Figure 7. It was found that six out of fourteen dimensions were within α < 0.05 for both significance tests; all six were within beta and gamma bands (25–100 Hz). 

It is worth pointing out that the per-dimension *p*-values shown in Figure 7 were computed independently, however, each multi-dimensional feature vector proposed in this work was considered as one single attempt during the classification. Such a scenario forms a typical multiple testing problem [46]—each dimension of the feature represented one attribute, the combination of the attributes (i.e., concatenation of the feature dimensions to form a vector) could potentially increase the diversity between the HC and MCI classes, and lead the *p*-value to be positively biased. Therefore, in order to battle against such overly optimistic inference, we adopted the Holm–Bonferroni method [47] to adjust the *p*-values. The detailed *p*-values and their corrected values for the six frequency ranges with α < 0.05 are listed in Table 3. Though most of the corrected *p*-values for the selected features were found to have increased after the correction, they were still well below the threshold of α = 0.05. The mean *p*-values of the six dimensions for the *t*-test and Kruskal–Wallis test were 0.01 and 0.02, respectively. These results suggest that the effective feature vector should contain at least six dimensions, which demonstrated statistical significance for both the *t*-test and Kruskal–Wallis test. It is worth noting that while training with only the features with *p*-value < 0.05 for both the *t*-test and Kruskal–Wallis tests (shown in Figure 7), the recognition accuracy of only 69.7% was obtained, which indicates that some of the excluded still can contribute to the classification. 

Hence, we propose to further optimize the neuromarkers by performing a standard principal component analysis (PCA) to reduce the dimensionality and leverage the useful information from multiple dimensions of the feature vector [37]. A set of tests were performed, each time increasing the number of feature vector dimensions by one. The averaged LOOCV recognition performances are shown in Figure 8. Note that during the LOO cross-validation process, only the features used for training were processed by PCA, which might explain the phenomenon that the recognition rates in Figure 8 were found not to be monotonically increasing along with the addition of preserved dimensions. Interestingly, instead of six dimensions, Figure 8 shows that the average recognition rate reaches a maximum value when seven dimensions were kept after PCA. This was expected—though there were six dimensions satisfying the condition of α < 0.05 for both *t*-test and Kruskal–Wallis test, but a few other dimensions, such as those dimensions corresponding to 0–50 Hz, 75–87.5 Hz, and 87.5–100 Hz, demonstrated statistical significance with α < 0.05 for one of the tests (Figure 7). It appears that the features with these dimensions contributed positively to the HC vs. MCI classification after the PCA transformation was performed. Indeed, keeping six dimensions of the feature vector still provided the second-best recognition performance. Note that the results in Figure 8 show the performance using features from the hyper-COH group, with each ROI of that group generated feature vectors of 14 dimensions initially. The highest mean recognition rate of 83.8% was achieved after performing PCA, which surpassed the best performance of 81.64% reported in Section 3.2 Figure 4, where the features with all the 14-dimensions were kept for classification. 

## 4. Discussion and Conclusions

This work explored the functional connectivity (FC) changes across brain regions (regions of interest, ROIs) between participants with mild cognitive impairment (MCI) and healthy control (HC) subjects. The findings, based on MS-COH indicators, were then verified using a wavelet-based source-space neuromarker, tested through a MEG database containing 40 participants (20 MCIs and 20 HCs). We identified two distinct ROI groups: one with hyperconnected FCs and another with hypoconnected FCs. The proposed neuromarkers obtained using the data from the hyperconnectivity ROI group provided much-improved recognition performance compared to that using the hypoconnectivity ROI group for MCI detection. We also found that when the features from all the 10 ROIs (both hyper and hypo connected regions) were concatenated for classifier training, the recognition performance could not be improved further—the average accuracy rate was practically degraded to 74.72%, which was about 10% worse than using only the data from hyper-connectivity regions for the classification. 

Some of the findings from this work may be considered to extend the existing understanding of the functional roles of the selected ROIs. We noticed that the left-brain half (hemisphere) tends to be more effective in MCI detection: out of five ROIs, four of them demonstrated hyper-connectivity (LF, LT, LP, and LO), whereas, in the right half, only two ROIs (RC and RT) showed such phenomenon. It is well-recognized that the left hemisphere of the cerebrum normally controls speech, comprehension, arithmetic, and writing; the right half controls creativity, spatial ability, artistic, and musical skills [48]. In this study, the hyper-connectivity of the left half may, therefore, indicate cognitive decline, which is reflected in the observed speech and comprehension difficulties of the elder MCI participants. The regions of interest that we found to be more discriminating for MCI detection are also somewhat in line with other published work [49], where, for subjects with mild cognitive impairment and Alzheimer’s disease, diminished connectivity as degeneration progresses has been observed based on fMRI modality, whereas for subjects with traumatic brain injury and multiple sclerosis, increased connectivity was observed. Hyperconnectivity is a common response to neurological disruption and that it may be differentially observable across brain regions. 

Based on the proposed implementation, our best mean MCI detection rate is 83.8% using the LOO cross-validation. A considerable difference in MCI detection performance was also noticed between using the MCR and LOO cross-validation approaches while using the same feature for classification. Such a difference deserves some further consideration. It is reported that the state-of-the-art MEG features have already been quite effective for MCI detection (more than 98% of recognition accuracy [8]), based on some conventional MCR cross-validation experimental schemes. However, when the test procedure imitated a real-world clinical scenario using individual participants (based on LOO cross-validation), the proposed neuromarkers were not reliable enough for MCI detection (with performances less than 80%). For example, the classification rate of 68.9% obtained using wavelet-based features [10] indicates that the need to develop more effective neuromarkers for MCI detection persists.

It is, therefore, recommended that features and classification schemes should be tested in real-world clinical settings, in such a way that fair comparisons can be made amongst various neuromarkers. As the best performance was obtained after the PCA transform, additional work could be conducted to resolve one limitation that remains in this work: the proposed feature set spans across multiple frequency bands with different resolutions. It will be interesting to explore and find the most effective feature combination for MCI detection, as well as its clinical origin, so that further improvement of the recognition performance may be achieved. As indicated in our previous work [50,51], we also suggest the inclusion of additional information, such as cognitive and functional assessment scores, along with other biomarkers, to improve the accuracy for MCI classification. Further, longitudinal studies should be included to distinguish MCI leading to AD cases from MCI to non-AD cases. To better facilitate the early detection and treatment, we also plan to include patients with Alzheimer’s disease for three-class classification in our future work.

## Figures and Tables

**Figure 1 sensors-21-06210-f001:**
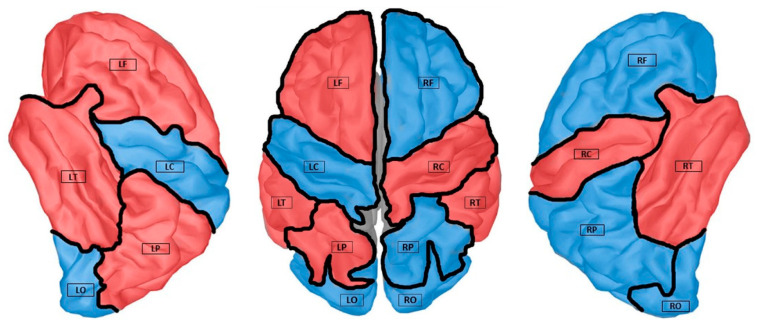
ROI partitions are generated using Brainstorm [21], modified from the standard Desikan–Killiany atlas parcellation and region labels description [22]. The three views show the ROI partitions of the cortical surface and the COH connectivity: red regions indicate the hyper-connection group while the blue regions indicate the hypo-connection group for MCI with respect to HC.

**Figure 2 sensors-21-06210-f002:**
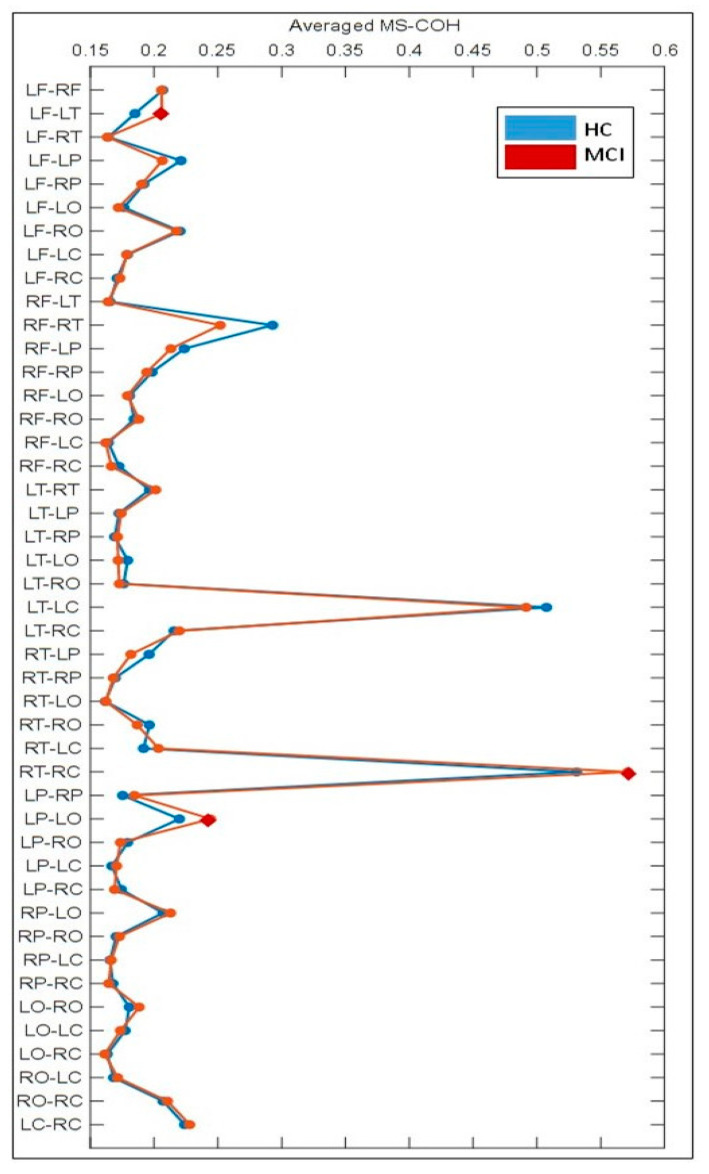
Comparison of average COHs for all regional pairs between HC and MCI subjects. LF-LT, RT-RC, and LP-LO pairs are highlighted using red diamonds. The ROI pairs of which are found increased COHs for MCI subjects. LT-LC region pairs are also illustrated for their relatively high FC value (on average). Note for these computations, the data of multiple participants were concatenated for analysis, the individual differences were purposely ignored.

**Figure 3 sensors-21-06210-f003:**
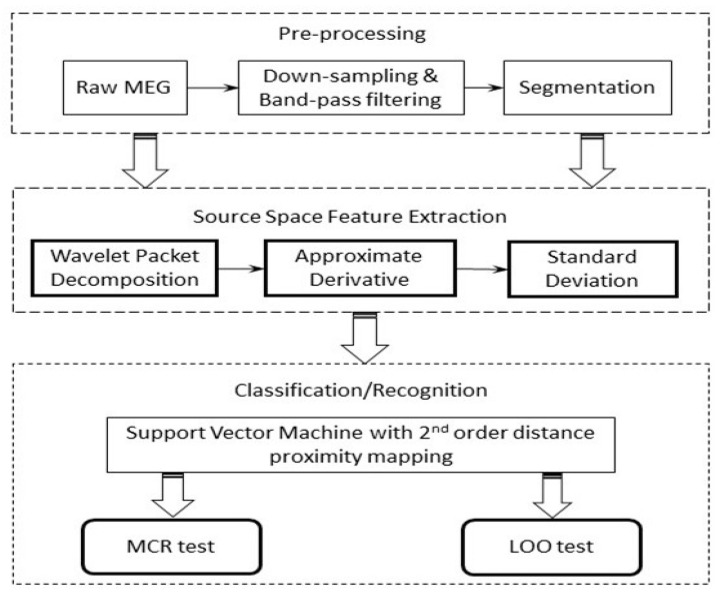
Workflow diagram of the proposed system.

**Figure 4 sensors-21-06210-f004:**
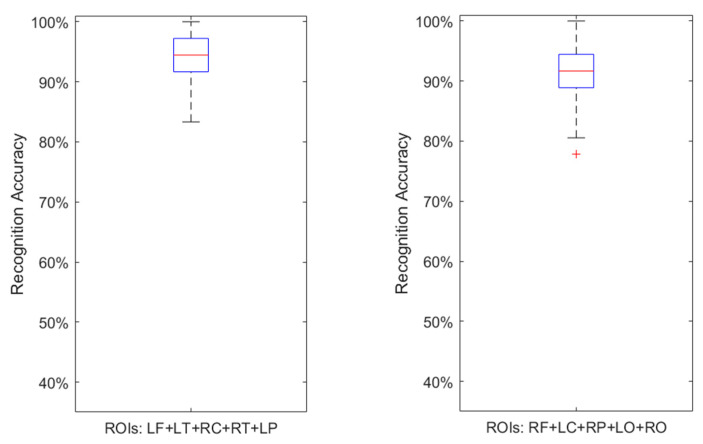
Boxplots for MCR approach using 3-NN. (**Left**): hyper-COH group, mean accuracy 93.83%; (**Right**): hypo-COH group, mean accuracy 91.39%. The red lines indicate the median accuracies.

**Figure 5 sensors-21-06210-f005:**
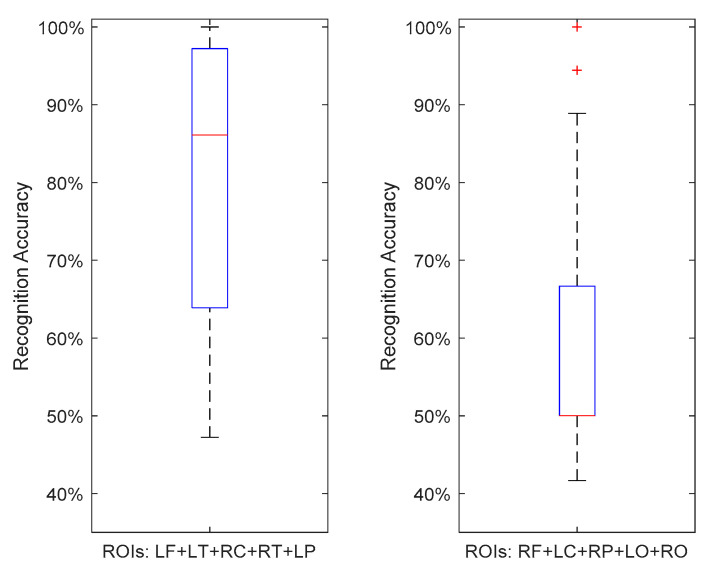
Boxplots for LOO approach using SVM. (**Left**): hyper-COH group, mean accuracy 81.64%; (**Right)**: hypo-COH group, mean accuracy 61.33%. Red lines indicate the median accuracies.

**Figure 6 sensors-21-06210-f006:**
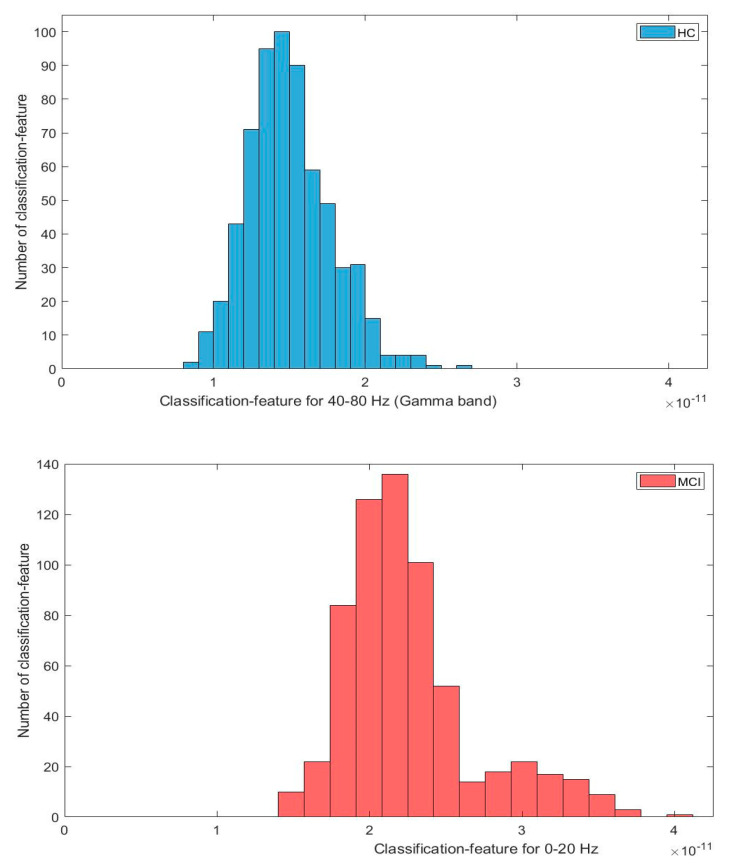
Distributions of the classification features of two frequency ranges. The HC and MCI distributions are set as the same range for comparison.

**Figure 7 sensors-21-06210-f007:**
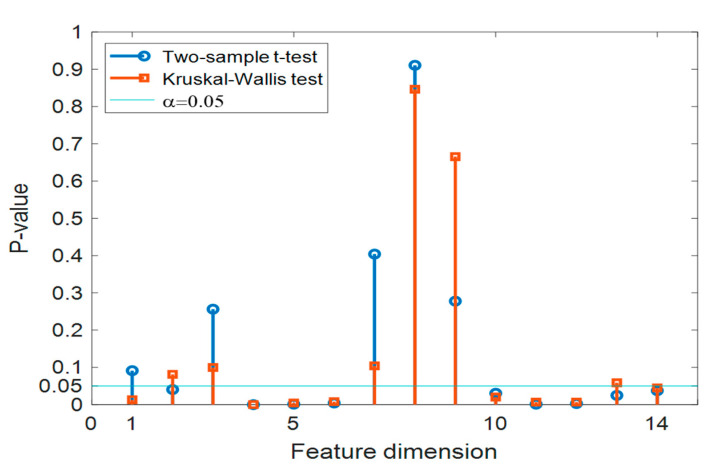
Statistical tests for the proposed wavelet feature, analyzed per dimension using two test approaches.

**Figure 8 sensors-21-06210-f008:**
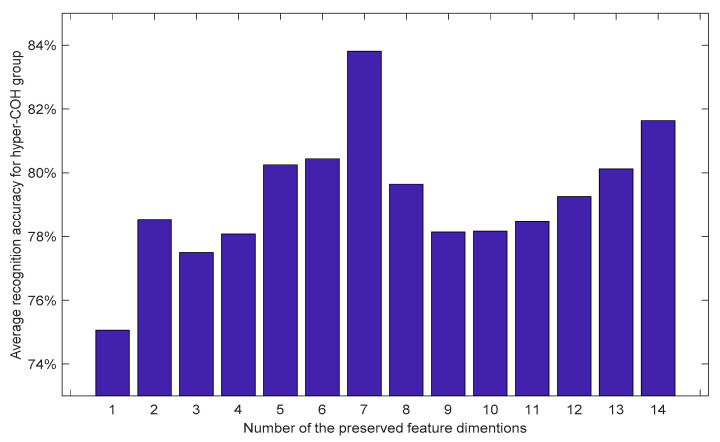
PCA analysis on the feature dimensions for the proposed classification features from hyper-COH group. The recognitions are the mean performance through SVM using the LOO approach.

**Table 1 sensors-21-06210-t001:** Statistical measures of FCs based on MS-COH for ten ROIs. Each ROI contains 3600 summed MS-COHs from the nine remaining ROIs. The highlighted (in bold) values relate to the hyper-connectivity of the regions for MCI subjects.

Stats.	Mean		Max		Min		SD	
ROI	HC	MCI	HC	MCI	HC	MCI	HC	MCI
**LF↑**	3.36	3.53	5.55	5.81	2.26	2.29	0.51	0.56
RF↓	3.54	3.53	9.49	7.87	1.91	2.17	0.80	0.66
LT↓	3.20	3.19	5.76	5.53	2.00	2.15	0.47	0.44
RT↓	3.65	3.53	7.40	6.22	2.13	2.25	0.67	0.58
LC↓	3.21	3.19	5.09	4.89	2.01	2.29	0.42	0.39
**RC↑**	3.25	3.26	5.34	5.30	2.23	2.05	0.46	0.44
LP↓	3.42	3.40	8.40	6.86	2.04	2.07	0.66	0.54
RP↓	3.29	3.26	8.76	5.55	2.16	2.20	0.62	0.49
LO↓	3.65	3.62	8.92	6.35	2.26	2.10	0.74	0.61
RO↓	3.32	3.31	5.65	5.24	2.00	2.12	0.51	0.47

**Table 2 sensors-21-06210-t002:** Effectiveness of ROI group selection in different testing protocols. The results are averaged from 20 tests in both scenarios.

ROI Group and Scenario	Mean Sensitivity	Mean Specificity	Balanced Accuracy
Hyper-COH ROI with MCR	92.72%	94.94%	93.83%
Hypo-COH ROI with MCR	91.44%	91.33%	91.39%
Hyper-COH ROI with LOO	86.61%	76.67%	81.64%
Hypo-COH ROI with LOO	32.06%	90.61%	61.33%

**Table 3 sensors-21-06210-t003:** The *p*-values and their Holm–Bonferroni corrections for both significance tests. Only the bands (dimensions) with α < 0.05 are listed.

Stats.	Two-Sample *t*-Test		Kruskal–Wallis Test	
Band (Hz)	*p*-Value	Corrected *p*	*p*-Value	Corrected *p*
25–50	$5.4\text{×}10^{-5}$	$3.2\text{×}10^{-4}$	$7.1\text{×}10^{-6}$	$4.2\text{×}10^{-5}$
37.5–50	0.0306	0.0306	0.0203	0.0203
50–75	0.0014	0.0062	0.0043	0.0214
50–62.5	0.0012	0.0062	0.0066	0.0266
62.5–75	0.0029	0.0088	0.0068	0.0266
75–100	0.0045	0.0089	0.0080	0.0204

## Data Availability

The data for this study is not publicly available.

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
