# Peer review of "Detection of Mild Cognitive Impairment with MEG Functional Connectivity Using Wavelet-Based Neuromarkers"

_sensors, 2021, doi:10.3390/s21186210_

Round 1
Reviewer 1 Report
The paper proposes using wavelet-based source-level neuromarkers for mild cognitive decline detection based on magnetoencephalographic (MEG) signals. The paper is well prepared. The methodology is sound, and the technical presentation is good. I have only a few comments:
- Present an outline of your methodology as a figure (workflow diagram).
- Line 222: present also formulas for the calculation of auto-spectral density.
- Present a motivation for the selection of classifiers (kNN, LDA, SVM). These are old, well-known classifiers. Why did you not consider a neural network for classification?
- Present a more detailed description of the approximate derivative procedure.
- Figure 6: why is the ROC curve of Hypo-COH ROI with LOO so much worse than random? Table 2 does not support such evaluation.
- Discuss the limitations of the proposed neuromarkers.
Author Response
Thank you very much for the comments, please find the attached file for the responses & updates for the suggestions.

Reviewer 2 Report
In this manuscript, the authors report findings from applying machine learning based techniques to distinguish among participants with mild cognitive impairment (MCI) and normally aging healthy controls (HC) based on connectivity patterns among different brain regions using data derived from applying wavelet analysis to source-localized magnetoencephalography (MEG) data. The application of machine learning techniques to neuroimaging data is an area of increasing interest, and this contribution adds to that body of work. While the work is of sufficient quality overall, certain modifications are needed prior to considerations of acceptance to further strengthen the work.
Major concerns:
- While the authors mention various studies in the Introduction, it is unclear how these lead to their specific study or hypotheses. For example, the authors state that "As seen in the aforementioned works, ML techniques have shown a high variability in MCI detection performance." in the paragraph introducing the objectives of their study but then do not address this issue at all in their stated objectives. If the stated gap is that other studies have high variance - does your work have lower variance or did you explore the causes of variance? If not, perhaps rephrasing the Introduction and referencing other works that better lead to the gaps (and hypotheses) that authors are addressing will be better.
- The organization of the paper needs to be improved. As it stands now, the authors have combined their methods and results within the different analysis sections and this makes it quite confusing to read. The authors should reorganize the manuscript by having a Methods section and a Results section (with subsections for the various analyses that they would like to showcase).
- Some of the justifications for the methods are not clear. For example, on page 5 line 212, the authors mention that functional connectivity using broadband EEG has been effective for seizure detection, and so was applied to the current work. How does seizure detection relate to MCI detection? Are there commonalities among the two? If so, state the reasoning else better justify the techniques used.
- The vast majority of the prior works that the authors cited for MCI detection using MEG utilized the Beamformer approach for source localization. However, the authors choose to use the MNE approach due to the inability of certain Beamformer techniques to localize concurrent activity at multiple sites. However, MNE itself also has drawbacks in terms of generating source localization patterns that are biased towards the cortical surface and creating more 'diffuse' activation patterns. Did the authors account for this? If not, why not also confirm the findings using Beamformer to ensure that the shortcomings of MNE are not impacting the FC values especially among adjacent parcellations?
- The authors undertake a series of t-tests (page 9, lines 332-337), but this assessment should instead use a ANOVA with 'region' as a factor followed by post-hoc t-tests.
- Similarly, on page 13 the authors mention that they used two-sample t-tests and Kruskal-Wallis in cases of non-normality in data. However, the correct non-parametric counterpart of two sample t-tests is the Mann Whitney U or Wilcoxon. Could the authors please clarify?
- The Discussion section needs to be further expanded - how do the variety of results fit into the existing knowledge base? The authors provide a wide array of results, and then in the discussion only focus on very small part of their findings in terms of classification results, and even then do not expand on the findings a lot. For example, they specify that their findings are in line with prior works (reference 47), but do not provide any additional details of what parts are in agreement and what parts may be in disagreement. Similarly, they authors provide a very high-level and vague description of hemispheric difference (lines 547-550) but do not link them to other neuroimaging findings in MCI patients.
Minor Concerns:
- Page 8, line 314-326: the authors undertake normality assessments of the data prior to application of the t-test. Since the t-tests are applied for the inter- and intra-hemisperic groups separately, the normality assessments should also be undertaken in the same manner rather than by combining the data as the authors have done. Additionally, if the authors are using a quantitative assessment of normality, the histograms are not needed and largely add clutter rather than adding value in the manuscript.
- On a related note, the authors should state the results of their statistical testing in full (page 9, lines 333-337) rather than simply stating the percentage of tests that passed the alpha threshold.
- It is confusing why the authors mention 20 subjects in their description of the machine learning techniques (page 11) when they collected data from 20 HC and 20 MCI - shouldn't they have 40 participants as the starting pool prior to subselection? The reorganization of the manuscript (see major concern #2) may also help with this issue by enabling the reader to keep track of the various analysis parameters.
- The box plot of classification results suggest that in some cases outliers may be influencing the mean (e.g. the median is near 50% or chance-level, whereas mean is at ~60%). The authors should expand on this in their manuscript and discussion.
Author Response

(The authors gave the same response as above.)

Round 2
Reviewer 2 Report
I want to thank the authors for addressing some of the concerns raised in the previous round of reviews. However, several are still outstanding and some author answers have raised additional concerns. Please see below for details:
- The authors state "The motive to introduce various works in the literature is to show that different approaches have been used to evaluate such recognition system, and such inconsistence of the evaluation metrics making the performances of the system hard to be justified. We are trying to emphasize that only the metric simulates the real-world scenario best should be adopted, by showing the difference between two mainstream approaches. " in their response to Major Concern #1 from the previous round. It is somewhat ambiguous what the two mainstream approaches they are referring to, but I'm assuming they are referencing the x-fold cross-validation vs. leave-one-out approach to classifier assessment. If that is indeed the case then it is unclear why they mention "...which of these evaluation metrics could reflect the real world scenario..." in their manuscript text - isn't it only the leave-one-out option that reflects the real world scenario of having a trained classifier and then being evaluated on never-seen-before data? Also, I'm not sure how the previous studies that have shown a wide range of variability in terms of results influence this question. The authors explicitly state the classifier assessment approach for only two of the studies cited (Dimitriadis et. al. and Poza et. al.) and indeed the study by Dimitriadis et. al. employed and compared both cross-validation and leave-one-out approaches. Furthermore, the variability in the prior works (which is a major stated reason for the authors to undertake the current study) could easily have been due to the very different measures (e.g. turbulence coefficient vs. CorrEnv) used in the prior works rather than due to the classifier performance quantification approach (i.e. x-fold cross-validation vs. leave-one-out). As stated in the previous round of reviews, the authors need to make a clear case for why their study is required and how it fits within the existing body of knowledge. If the stated goal is explanation of variability, then other possible factors for variability need to be examined. Alternatively, the authors could restate their goal as evaluation to examine if the cross-validation vs. leave-one-out differences observed by Dimitriatris et. al. are also present using different measurements (i.e. coherence as used in the current study instead of CorrEnv or PLV used by Dimitrias et. al.).
- Thank you for updating the section titles. However, the fundamental problem of technique descriptions and results being stated together remains. If the authors do want to keep the existing divisions/parts, please create sub-sections for the methods and results sections for each of the 2 major parts of the paper.
- The authors state in their response that "From signal process and machine learning perspective the Seizure and MCI detections are to some extents similar: both are based on EEG signal in this case, also capturing information from same frequency range. ". Unfortunately, there are two issues with this - a) the current study uses MEG and not EEG, and b) the features of importance for seizure detection may be quite distinct from MCI given the high-frequency inter-ictal spike patterns associated with seizures and the lack of them in MCI. Nonetheless, if the authors want to continue to use that as justification of their methods, they can do so, although it appears to be a missed opportunity to not better explain their methods in the context of their current work.
- The authors state in their response that they did undertake Beamformer vs. MNE comparisons. Please include this (and corresponding results) in the manuscript or at least as Supplementary Material as this helps strengthen the manuscript.
- Statistical testing issue #1: In the previous round of the review I had stated that "The authors undertake a series of t-tests (page 9, lines 332-337), but this assessment should instead use a ANOVA with 'region' as a factor followed by post-hoc t-tests.". The authors did not address this point at all - please do so as the pairs are not truly independent and therefore should not be assessed using paired t-tests prior to an omnibus assessment. Additionally, this also relates to a concern that has arisen due to one of the authors' responses. In their response document, the authors state that they undertook 3.24X10^5 paired t-tests - were any corrections applied (e.g. FDR or Bonferroni)? Please clarify.
- Statistical testing issue #2: In their response, the authors state that they did not employ the Mann Whitney U/Wilcoxon test, but rather relied upon the Kruskal-Wallis as they considered the two to be equivalent. However, when working with small samples as is the case in this study, certain statistical packages (e.g. R) implement and utilize exact null distributions for small samples for Mann-Whitney but not Kruskal-Wallis thereby potentially leading to incorrect statistical outcomes. Similarly, several packages implement continuity correction for Mann-Whitney but not Kruskal-Wallis. Therefore it is generally better to use Mann-Whitney as the direct non-parametric analogue of paired t-tests.
- Thank you for adding the clarifications and additional information in the other sections.
Author Response
Thank you very much for providing the additional suggestions on updating the manuscript. We have now modified/changed the paper at our best efforts given a short period of time allowed. Please find the attachment for the detailed response letter.
Best regards,
All authors
